# A Method of Population Spatialization Considering Parametric Spatial Stationarity: Case Study of the Southwestern Area of China

**Junnan Xiong [1,2], Kun Li [1,\*], Weiming Cheng [2,3,4] ![ID], Chongchong Ye [1] and Hao Zhang [1,5] ![ID]**

[1] School of Civil Engineering and Architecture, Southwest Petroleum University, Chengdu 610500, China; xiongjn@swpu.edu.cn (J.X.); 201822000599@stu.swpu.edu.cn (C.Y.); zhanghao@imde.ac.cn (H.Z.)
[2] State Key Laboratory of Resources and Environmental Information System, Institute of Geographic Sciences and Natural Resources Research, CAS, Beijing 100101, China; chengwm@lreis.ac.cn
[3] University of Chinese Academy of Sciences, Beijing 100049, China
[4] Jiangsu Center for Collaborative Innovation in Geographical Information Resource Development and Application, Nanjing 210023, China
[5] Institute of Mountain Disasters and Environment, CAS, Chengdu 610041, China
[\*] Correspondence: 201822000586@stu.swpu.edu.cn; Tel.: +86-028-83037604

**Abstract:** Population is a crucial basis for the study of sociology, geography, environmental studies, and other disciplines; accurate estimates of population are of great significance for many countries. Many studies have developed population spatialization methods. However, little attention has been paid to the differential treatment of the spatial stationarity and non-stationarity of variables. Based on a semi-parametric, geographically weighted regression model (s-GWR), this paper attempts to construct a novel, precise population spatialization method considering parametric stationarity to enhance spatialization accuracy; the southwestern area of China is used as the study area for comparison and validation. In this study, the night-time light and land use data were integrated as weighting factors to establish the population model; based on the analysis of variables characteristics, the method uses an s-GWR model to deal with the spatial stationarity of variables and reduce regional errors. Finally, the spatial distribution of the population (SSDP) of the study area in 2010 was obtained. When assessed against the traditional regression models, the model that considers parametric stationarity is more accurate than the models without it. Furthermore, the comparison with three commonly-used population grids reveals that the SSDP has a percentage error close to zero at the county level, while at the township level, the mean relative error of SSDP is 33.63%, and that is >15% better than other population grids. Thus, this study suggests that the proposed method can produce a more accurate population distribution.

**Keywords:** population spatialization; spatial stationarity; geographically weighted regression; DMSP/OLS; land use

## 1. Introduction

The study of populations, which straddles many fields, such as the environment, social development, and economics [1–4], is one of the core elements affecting sustainable development in today's world [5]. According to one projection, the world's population was estimated to have reached 7.7 billion people as of April 2019 [6]; however, population growth has been accompanied by numerous problems related to humanitarian issues, disaster planning, and assistance with economic development. Available and accurate spatial population distribution data are useful ways to analyze these types of problems [7]. Moreover, with the inclusion of population distribution, policies made by

governments could be made more rational by considering the sustainable development of resources and environmental protection [8], especially for populous countries. Therefore, the acquisition of spatial pattern data related to population distribution is crucial and necessary.

Demographic data provide the main basis of general information related to population distribution and composition [5]. In a large part of the world, authoritative demographical data are aggregated by spatial or administrative units [9] at regular intervals (once a decade in China). Generally, with the relatively coarse statistical analysis and limited types of data that are typically collected [10], demographic data reveal entire population counts in entire administrative units, instead of providing detailed spatial population distribution characteristics [11]. Therefore, these data cannot satisfy the need for practical applications that link these data to their geographical distribution. However, gridded population distribution datasets gained by spatialization can remedy these defects. Moreover, these datasets can be flexibly integrated with other, finer spatial datasets and summarized at any specified level of aggregation [12]. As supplements and substitutes of demographic data, gridded population distribution grids datasets can promote the development of population-related studies.

Previous studies have applied remote sensing data extensively to model population dynamics [13–16]. Night-time light data, especially from the defense meteorological satellite program's (DMSP's) operational linescan system (OLS) [17], have been widely used for large-scale population spatialization studies [18]. The free DMSP/OLS data have been proven to have excellent applicability to the observation of population dynamics [1,17,19,20]. However, these data are unreliable for use in spatialization when only using the original night-time light data [11]. To improve performance, many studies have frequently combined DMSP/OLS data with auxiliary data [9,21–25]. When choosing auxiliary data, researchers should consider improvements in simulation accuracy and avoid invalid composition models. Currently, land use data are commonly used with night-time light data in large-scale population spatialization studies; this has proven to provide a better performance than the use of single night-time light data alone [26]. Briggs [24] proposed four strategies that were based on land classification to conduct a regression analysis on light emissions, lit areas, and unlit areas of each land use type in each region and then gained 200 m and 1 km spatial resolution population density maps. In the present study, land use data were used to prevent the incorrect assignment of night-time light and to derive population distribution weights. Generally, the use of night-time light data makes it easier to distinguish an urban population, and attention is needed to accurately reflect the presence of a scattered rural population, especially in developing countries with large populations.

To obtain a population grid, models based on mathematical relationships established between demographic data and spatial variables are popular and reliable [27]. Therefore, based on the use of stable data, it is important to select and improve mathematical methods to produce very accurate results [10]. Commonly used methods mainly include dasymetric mapping methods [28,29], spatial regression models [30,31], and multi-source data fusion methods [18,32]. These methods have been used to construct several typical and widely used population distribution datasets, such as the Gridded Population of the World (GPW) [33], LandScan [34], the WorldPop project (WorldPop) [35], and the China Gridded Population Datasets (CGPD) [36]. However, some problems of models limit the accuracy and precision of the modeled results. These widely-used methods usually do not fully consider the parametric spatial stationarity (e.g., the population density weights determined by the same kinds of lands may be fixed or may vary with different positions), which may introduce a misallocation of population weights. In large-scale population spatialization studies, regression models are widely used, currently [37]. Many studies have redistributed population based on traditional global models, such as the ordinary least-squares (OLS) model, which presumes that the all relationships between the parameters and the results are stationary, resulting in an "average" behavior between the estimated parameters [38]. High rates of error usually result for areas with weak relationships in the OLS model [39]. Some studies have employed geographically weighted regression (GWR) [11], the most widely used method that allows all parameters to change geographically (or to be non-stationary). Although results indicate that the accuracy of GWR-based models is higher than that of OLS-based

models, variables affect populations with different spatial patterns, and some of them may not vary, while others have global effects. In addition, numerous research studies have used zoning to enhance the model accuracy. Sutton [40] employed the economic level as a zoning criterion. Cheng [41] conducted zoning research according to the population size and the level of urbanization in counties. Zeng [26] used night-time imagery clustering and a shortest path algorithm to create eight zones. However, most of the zoning methods have divided a study area into very few zones, although inter-zonal differences were emphasized; no method has revealed the differences in the spatial distribution of a population within a zone [37]. The enhanced effect is not very significant. Furthermore, methods that employ suitable zones and assign weights still have limited validity and versatility when used with population modeling, especially for regions with a scarce amount of data (or complex environments). In this study, we used the term "traditional" to describe these methods which cannot differentiate between the spatial stationarity and non-stationarity of variables. Suitably estimating the population density of different distribution patterns under different regions is still a major limitation of generally-used population spatialization methods. Moreover, the population distribution datasets derived from these traditional methods face the awkwardness of an even population density occurring inside individual regions. In recent studies, models that consider the spatial stationarity of variables have usually performed better than other traditional models [39,42]. Considering spatial stationarity, sufficiency may also be an important factor in further enhancing the accuracy of a population model. Therefore, in the present study, spatial stationarity was incorporated in a population model to differentiate among various spatial patterns with the goal of improving the model's accuracy.

Based on the above arguments, our study aimed to obtain suitable and accurate estimates of populations with a novel population spatialization method which integrates demographic, night-time light, and land use data. The method uses an s-GWR-based model which mixes parametric stationarity and non-stationarity to enhance the precision of population spatialization. The southwestern area of China was selected as the study area for verification. In addition, we analyzed the effects of the variability of variables in the modeled population by comparing global (OLS), local (GWR), and mixed (s-GWR) modeling approaches in the study area. Subsequently, we derived a spatial distribution of the population (SSDP) using an s-GWR model of the study area in 2010 for effectiveness assessments. This study may help to avoid the concealment of the local heterogeneities between regions when compared with traditional global and local models, and it provides a fast and accurate method to generate gridded population data. Moreover, the comparison with previous population datasets attempts to identify any flaw that affects the accuracy of the analysis, which, in turn, can assist in reasonably improving future population distribution modeling attempts.

## 2. Study Area and Datasets

### 2.1. Study Area

The study area in southwestern China includes all or part of four provinces: Yunnan, Sichuan, Guizhou, and Chongqing. This study area spans from 21°08' N to 34°18' N and 97°20' E to 110°11' E (Figure 1) while covering an area of 1,127,800 km². Southwestern China has a complex terrain, which can be divided into three terrain units, including the Sichuan Basin with surrounding mountainous regions and hilly regions of the Yunnan–Guizhou Plateau, and the alpine regions of the Qinghai–Tibet Plateau [43]. Meanwhile, due to the unbalanced economic development in the study area, there are more economically developed regions mainly distributed over the eastern region with relatively flat terrain, such as the Chengdu Plain. The region has 46 municipal boroughs and 435 county-level administrative regions. According to census statistics, in 2010, the study area's population was close to 190 million (including 76 million urban residents and a rural population of 117 million), accounting for 14.18% of the total population. The population is distributed unevenly across these regions due to the differences in economy and geography, and it is more densely populated in the east than in the west [44]. It is noticeable, however, that southwestern China includes one of the main areas targeted in

China's Develop-the-West strategy and serves as an important maritime export region of the maritime silk road under China's Belt and Road Initiative. Therefore, this region is an ideal study area on account of the diverse population density and its important role in China.

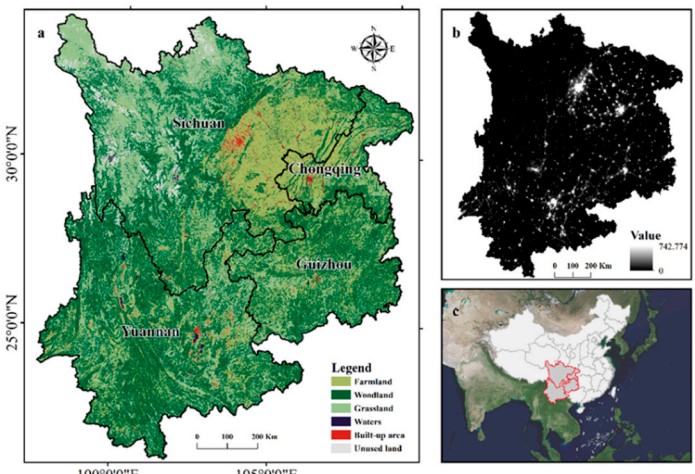

**Figure 1.** Study area of four southwestern provinces in China: (**a**) Land use patterns; (**b**) night-time light imagery in 2010; and (**c**) location of the study area in China.

## 2.2. Data Sets and Pre-Processing

### 2.2.1. Night-Time Light Imagery

In this study, we used radiance-calibrated DMSP/OLS imagery from 2010 with a 30 arc-second resolution, which was provided by the U.S. National Geophysical Data Center (NGDC) (https://ngdc.noaa.gov/eog/dmsp/download_radcal.html (accessed 1 May 2019)). The original stable DMSP/OLS data experienced "blooming" and "overflowing". In addition, despite having a great number of research achievements based on the saturation correction of stable DMSP/OLS data [45,46], some uncertainty still exists [9,22]. This product was developed to solve the saturation effect, which exists in the stable DMSP/OLS night-time light product by merging the imagery collected at different gain settings; it also contains a global radiation corrected imagery time of the series 1996–2010. Moreover, the Digital Number (DN) values of stable DMSP/OLS night-time light range from 0 to 63, but the radiance-calibrated product does not have this restriction, making it an ideal dataset for this method of research. Albers Conical Equal Area projection was applied to the light data, and the re-projected map was resampled to 1 km using a nearest-neighbor resampling algorithm; then, data were masked by the boundary of the study area. Figure 1b shows the night-time light imagery after processing.

### 2.2.2. Land Use Data

The land use data employed here came from the National Land Use/Cover Database of China (NLUD-C) produced by the Chinese Academy of Sciences. It contained data on land use/cover in China for five periods (including the 1980s, 1995, 2000, 2005, and 2008) [47–51] and was updated in 2010 [52]. The original data were produced at a 1:100,000 scale and fell into six land use categories and 25 sub-categories based on a hierarchical classification system. The NLUD-C has been the most accurate (more than 95.41%) and authoritative dataset for the study of land use in China for years.

To maximize the accuracy of the original data and unify resolution, we converted 25 land use sub-categories to 25 raster files with a 1 km resolution using the fishnet tool in ArcGIS (ESRI, Inc., Redlands, CA, USA) software. The area of each land cover type in each cell was calculated as the percentage of cell area as a pixel value; thus, each raster layer corresponded to a particular land cover type. Figure 1a shows the distribution of land use in the study area.

2.2.3. Population Data and Boundary Map

China conducts a census every ten years. There have been six nationwide population censuses: 1953, 1964, 1982, 1990, 2000, and 2010. In this study, the original 2010 census data were obtained from the State Statistical Bureau. County and township-level administrative boundary maps (scale of 1:4,000,000) were provided by the Institute of Geographic Sciences and Natural Resources Research of the Chinese Academy of Sciences. Due to the influence of digitization errors and other factors, such as the boundary changes of administrative regions, the boundaries of administrative units did not completely match with census data. After necessary adjustment, ArcGIS software was used to correlate attribute data with the corresponding spatial data of administrative units; eventually, valid data were obtained from 435 counties and 6157 towns.

For rational verification of the experimental results, the three latest and widely used population distribution datasets from 2010 were also used in this study: GPWv4, LandScan, and CGPD. All data were masked by the boundary of the study area and re-projected to Albers Conical Equal Area projection using a bilinear resampling algorithm with a 1 km spatial resolution. It was noticeable that the bilinear resampling algorithm used four known pixel values around the sampled point to participate in the calculation, and it could do some smoothing on the data. If there was no pixel on one or both sides, the edge or linear information was lost. Table 1 shows the details of the data used in this study.

**Table 1.** List of datasets and sources.

| Data Type | Resolution | Source |
| --- | --- | --- |
| DMSP/OLS night-time light | 30 s | National Geophysical Data Center, USA |
| Land use/cover data | 1:100,000 | National Land Use/cover Database, CHN |
| Census data | County and township levels | State Statistical Bureau, CHN |
| Boundary | County and township levels | Chinese Academy of Sciences, CHN |
| GPWv4 | 30 s | Center for International Earth Science Information Network, USA |
| LandScan | 30 s | Oak Ridge National Laboratory, USA |
| CGPD | 1 km | Resources and Environmental Sciences Data Center, CHN |

## 3. Method and Modeling

The spatialization of a population is a spatial disaggregation process where census data are downscaled to a grid with auxiliary data and statistical techniques [53]. In this paper, we employed Pearson's correlation test to verify the statuses of land use types, which can reflect the range of population activities suitably. Then, DMSP/OLS data were integrated with the selected land types to extract independent variables, aiming to reflect inter and intra-class differences. After conducting a statistical analysis on these parameters for each county, we used a geographic variability test to distinguish among spatial patterns of variables (global or local) to introduce the incorporating the spatial stationarity of variables into the population model. Based on the results, an SSDP was generated by an s-GWR model with the selected variables. Finally, an accuracy assessment that contrasted with previous studies was applied on the SSDP. Figure 2 shows the key steps involved in modeling the population in this study.

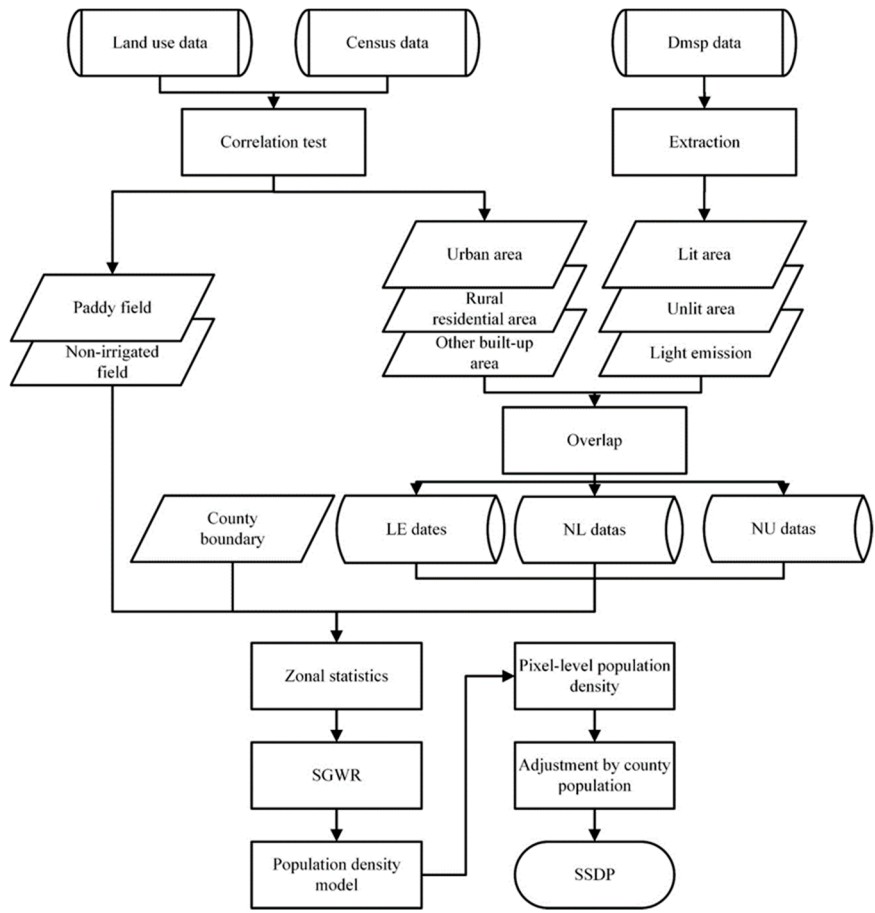

**Figure 2.** Flowchart for modeling the population in this study.

## 3.1. Extracting Independent Variables

### 3.1.1. Correlation Test between Population and Land Use

In population disaggregation, a population should not be evenly distributed across each kind of land use. Therefore, Pearson's correlation coefficient (PCC) was applied to select proper land types that were associated with the population. In the process of modeling a population with land use data, considering the actual situation of population distribution, the water and unused land types do not participate in the analysis of spatialization. Next, we summarized the area data of the 12 remaining land sub-categories and census data in each county with zonal statistics by using ArcGIS software; then, the correlation coefficient was calculated between land types and population by using SPSS (IBM Corp., Armonk, NY, USA) software.

The PCC is, in essence, a statistical method that can quantitatively measure the correlations between variables. Therefore, the PCC is usually treated as the criterion used for selecting variables. The formula for PCC is shown as follows:

$$r = \frac{N\sum x_i y_i - \sum x_i \sum y_i}{\sqrt{N\sum x_i^2 - (\sum x_i)^2}\sqrt{N\sum y_i^2 - (\sum y_i)^2}},\tag{1}$$

where $r$ is the value of the correlation coefficient; $x_i$ is the population count of county $i$; $y_i$ is the area of a certain land category in county $i$; and $N$ is the number of counties in the study area. The range of PCC is from −1 to 1. With an increase in the absolute value of PCC, the correlation between these two variables increases or adversely decreases when the absolute value approaches zero.

3.1.2. Integrating DMSP/OLS Data with Land Use Data

In this population spatialization model, the amount of night-time light observed should indicate different weights for the populations present in different land use types; the value of a population cannot be negative [26]. Thus, the land types shown to have a significant, positive *r* value in Section 3.1.1 were selected for extraction. With reference to previous methods, we first developed three raster layers from radiance calibrated in a DMSP/OLS image: light emission, lit areas, and unlit areas. Then, an overlay analysis was used to integrate these three layers with the selected land use types. Finally, we were able to obtain three datasets: light emission in pixels (LE), the number of lit pixels (NL), and the number of unlit pixels (NU). Note, according to the criterion of "non-residential area, no light", during the process of overlay analysis, night-light data needed to be related to the residential area. However, the population is not always distributed in the built-up areas, due to the accuracy flaws and display standard issue of map spots in land use products, which are based on satellite remote sensing. In other land types, scattered rural settlements may exist, such as isolated houses of farmers and herdsmen, tents, gers, and other facilities. The above-mentioned scattered but numerous residential facilities cannot be reflected in a 1:100,000 scale land use map, but they do exist. To reduce the possible underestimation of sparse populations in rural areas, populations in other land use types should not be ignored. Thus, the area information of additional land types should be included in a model. Therefore, we used the "Zonal Statistics" tool to summarize all independent variables at the county level for the spatialization process.

*3.2. Spatialization of Population*

The spatialization model can be described by the regression relationships between populations, land use types, and night-time light data. The model was built by simultaneously using the OLS and GWR models in addition to an s-GWR model. Three commonly used indicators of regression performance exist: the Akaike information criterion (AICc), determination coefficient ($R^2$), and adjusted $R^2$. The AICc is effective for verifying the goodness of fit [54]. According to the rule-of-thumb, a model with an AICc difference, which is greater than or equal to two, has an actual improvement in fitness [55]. To complete the process, GWR 4.0 [56] software was used.

The global linear relationship between a population and the extracted variables can be described quantitatively by an OLS regression model. The model can be expressed as follows:

$$y_i = \alpha_0 + \sum_{j=1}^{k} \alpha_j x_{ij} + \varepsilon_i, \qquad (2)$$

where $y_i$ is the estimated population; $k$ is the number of independent variables; $\alpha_j$ is the coefficient of the *j*th variable $x_{ij}$; $\varepsilon_i$ is the random error that meets spherical perturbation assumptions; and $\alpha_0$ is a constant that is set to zero.

The OLS model ignores the spatial non-stationarity of the coefficient. The GWR model is an extension of the general linear regression model and places emphasis on parametric non-stationarity [11]. The geographical locations of the data are embedded into the regression model; therefore, the regression coefficient becomes a function related to the spatial location. Additionally, the locally spatial relationship between the population and the extracted variables can be expressed by the GWR.

However, because of the different economic levels and living habits of the population in the study area, different population distribution patterns exist, so populations determined by the same variable may be fixed and varied depending on the location. Nevertheless, the semi-parametric, geographically weighted regression (s-GWR) model was proposed to cope with this situation, in which characteristics of global and local models were mixed and the fixed global variables were allowed to coexist with other variables that change geographically [57]. Generally, this mixed model has commonly been used for spatial relationship analysis and has performed better than other models

in recent studies [39,42]. Compared to simplex global or local methods, the s-GWR model realizes semi-parametric non-stationarity by combining globally fixed and geographically varying terms. Thus, this study attempts to introduce it to population estimates, and the population distribution can be expressed by s-GWR as follows:

$$y_i = \sum_{m=1}^{l} \alpha_m z_{im} + \sum_{j=1}^{k-l} \beta_j(u_i, v_i) x_{ij} + \varepsilon_i, \tag{3}$$

where $\alpha_m$ is the fixed coefficient of the $m$th global independent variable $z_m$, and $\beta_j(u_i, v_i)$ is the $j$th parameter estimate for the local variable $x_{ij}$ at each location $(u_i, v_i)$. To derive the mixed pattern of s-GWR, a geographical variability test, which is similar to stepwise regression, was applied in the present study to assess the variability between variables with the help of GWR4.0 software. Then, we were able to differentiate between the global and local patterns of variables. A geographical variability test is a kind of recursive model that uses comparisons to select variables as either fixed or varying terms under a criterion such as AICc. We compared the model where the $k$th variable was switched to a fixed term to the model treating all variables as variable terms to test the geographical variability of the $k$th varying coefficient. If the AICc of the switched model is smaller, it is best to assume that the $k$th variable is spatially stationary and the "Diff of Criterion" value for the variable appears positive. Otherwise, a larger AICc indicates that a spatially non-stationary relationship and a negative "Diff of Criterion" of selected variables exist. The 11 variables extracted were tested by this comparison routine using the same bandwidth. Furthermore, it was necessary to carry out a spatial autocorrelation test before constructing the GWR model; therefore, Moran's I tool in the ArcGIS software was used to analyze the spatial distribution pattern of the population.

The three models mentioned above were constructed at the county level and then applied to the pixel level. An adjustment had to be made to resize the estimated population to match with census data at the county level. The adjustment method can be expressed as follows:

$$y_{in}' = y_{in} \times \left( \frac{\overline{y_i}}{y_i} \right), \tag{4}$$

where $y_{in}'$ is the pixel-level population after adjustment; $y_{in}$ is the estimated population of the $n$th pixel in county $i$; $\overline{y_i}$ is the census data of the $i$th county; and $y_i$ is the total estimated population of a county.

### 3.3. Accuracy Assessment

It was necessary to carry out an accuracy assessment to analyze the applicability of gridded population production. In reference to the methods used to verify and validate simulation results in previous studies [5,9,11], except for the aforementioned indicators (such as $r$, $R^2$, and AICc), the mean error (ME), mean relative error (MRE), root mean square error (RMSE), and median percentage error (MPE) were applied for validation. They are defined as

$$ME = \frac{1}{n} \sum_{i=1}^{n} \frac{p_i - \overline{p_i}}{\overline{p_i}}, \tag{5}$$

$$MRE = \frac{1}{n} \sum_{i=1}^{n} \frac{|p_i - \overline{p_i}|}{\overline{p_i}}, \tag{6}$$

$$RMSE = \sqrt{\frac{\sum_{i=1}^{n} (p_i - \overline{p_i})^2}{n}}, \tag{7}$$

where $n$ is the number of statistical units; $p_i$ denotes the total estimated population in statistical units; and $\overline{p_i}$ is the actual population. In addition, MPE is equal to the 50th percentile of $n$ relative errors.

## 4. Results and Discussion

*4.1. The Results of the Population Spatialization Models*

4.1.1. The Extraction Results of Variables

The correlation coefficients ($r$) between the land use types and census data of the county level in the study area are presented in Table 2. In particular, $r$ values significant at the level of 95% in the $t$-test can be adopted. Therefore, the results show there are ten effective values; specifically, two sub-categories of farmland and three sub-categories of the built-up area have positive associations with the population, while others have negative ones.

**Table 2.** The correlation coefficient between land use sub-categories and the population.

| Paddy Field | Non-Irrigated Field | Forest | Shrub | Sparse Woodland | Other Woodland | Dense Grassland |
|---|---|---|---|---|---|---|
| 0.576 ** | 0.548 ** | −0.323 ** | −0.241 ** | −0.022 | 0.011 | −0.223 ** |
| **Moderately Dense Grassland** | **Sparse Grassland** | **Urban Area** | **Rural Residential Area** | **Other Built-Up Area** | **Waters** | **Unused Land** |
| −0.266 ** | −0.223 ** | 0.504 ** | 0.312 ** | 0.226 ** | – | – |

Notes: ** significance = 0.01.

According to previous work [11] and a correlation test, three sub-categories of built-up layers (urban areas, rural residential areas, and other built-up areas) were used to develop variables with three layers of DMSP/OLS data. Then, two farmland sub-categories (paddy fields and non-irrigated fields) were used to represent the weight of the probability of the spare population distribution. All in all, to model the population, we extracted eleven independent variables, as shown in Table 3: LE, NL, and NU for urban areas, rural residential areas, and other built-up areas, in addition to areas of paddy fields and non-irrigated fields.

**Table 3.** Summary of semi-parametric, geographically weighted regression model (s-GWR) models and results of a geographic variability test.

| Variable | | Min | Median | Maximum | DIFF of Criterion |
|---|---|---|---|---|---|
| Farmland | Paddy field | −801.74 | 358.8 | 1393.817 | −1.344 |
| | Non-irrigated field | 197.968 | 443.282 | 1143.394 | −81.126 |
| Urban area | LE | −326.563 | 21.798 | 548.14 | −48.856 |
| | NL | −19,447.847 | 12,331.588 | 95,261.748 | −3.5 |
| | NU | – | −368.957 | – | 6.272 * |
| Rural residential area | LE | −780.177 | −10.471 | 364.444 | 1 |
| | NL | −26,636.113 | 6978.858 | 97,907.33 | −2.757 |
| | NU | – | −2629.097 | – | 10.192 * |
| Other built-up area | LE | −1589.355 | 59.123 | 1072.292 | −13.763 |
| | NL | −116,595.952 | 535.835 | 98,971.153 | −2.281 |
| | NU | – | 3014.622 | – | 4.248 * |

Notes: * variables were changed to a global pattern.

### 4.1.2. The Spatialization Results

The Moran's I value of the population of a county was 0.451, which was processed at the 0.01 significance level. A positive Moran's I value stands for a positive spatial autocorrelation of the total population. Therefore, the distribution of the population shows a cluster tendency, which is an essential requirement of the geographical regression model. The spatialization of the population based on selected variables was constructed using OLS, GWR, and s-GWR models with GWR4.0 software. Table 4 summarizes the goodness of fit that was used to evaluate the performance of the regression accuracy, which included $R^2$, adjusted $R^2$, and AICc values. This indicates that the global regression model OLS can explain 78.3% of the variability of the population. Then, when the model was converted to a geographical regression model, considering the local effects of all eleven independent variables, the explanatory power increased to 89.2%, and the AICc value declined from 11,569 to 11,403 (Table 4); therefore, the model fit was significantly ameliorated. Table 3 shows the coefficients of selected variables obtained using an s-GWR model. Because of the possible interplay of a single land area's three factors (LE, NL, and NU), some of the coefficients were negative. We retained those factors, since they played a combined positive role in a pixel. Meanwhile, Table 3 shows the results of the "DIFF of Criterion", which was exported by a geographic variability test, and the positive NU values of the urban areas, rural residential areas, and other built-up areas indicated that they were not suitable to be spatially non-stationary. Therefore, these three variables were considered to be global, while the other eight variables switched to being considered as local terms in s-GWR. Finally, although the goodness of fit of OLS and GWR suggested that they can export rather good and reasonable results in the spatialization of a population, the s-GWR model explained 92.7% of the variance of the population, further reducing the AICc to 11,315; thereby, making the model fit better as a result of considering the spatially non-stationarity of variables. When compared with the above regression results, this indicated that the s-GWR model produced an improvement over the OLS and GWR models.

**Table 4.** Comparison of the models' fitting performances.

| Index | OLS | GWR | s-GWR |
|---|---|---|---|
| $R^2$ | 0.783 | 0.892 | 0.927 |
| Adjusted $R^2$ | 0.777 | 0.866 | 0.9 |
| AICc | 11,569.558 | 11,403.654 | 11,315.878 |

Figure 3 shows the spatial distributions and regional variations of the local $R^2$ values measured in the s-GWR and GWR models. The local $R^2$ had high values for the middle and northern areas and lower values in the southern areas. The local $R^2$ values of s-GWR of southwestern Sichuan and northern Yunnan showed an improvement over GWR; the local $R^2$ of >84% of the counties was higher than that of the GWR model, proving that the s-GWR model had a better performance than the traditional global and local models.

Based on land use, night-time light, and census data, we generated a 1 km spatial distribution of the population in the study area in 2010 using the s-GWR model (SSDP), which, when compared to the mean density by county, had the same general tendency as the population but provided more details (Figure 4). Furthermore, when contrasted with the land use and night-time light maps (Figure 1), the SSDP showed that the high population density area matched with the locations of lit residential land; conversely, the farmland areas or unlit areas corresponded to low population densities, which confirmed the distribution of the population in China. Additionally, the built-up areas carried the greatest population distribution weights, and sparse populations in rural areas were assigned to farmland. Lit residential land accounted for about 3% of the study area, where more than 41% of the total population lived, and the rest was accounted for by the wide, unlit area with a low population density. To a great extent, the SSDP constructed by census data using the s-GWR model matched the real distribution of the population spatially.

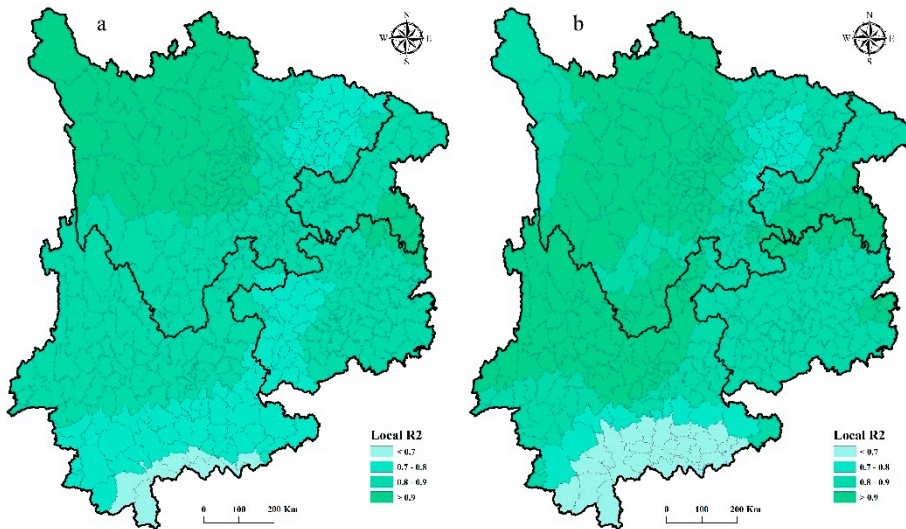

**Figure 3.** Maps of local $R^2$: (**a**) GWR; (**b**) s-GWR.

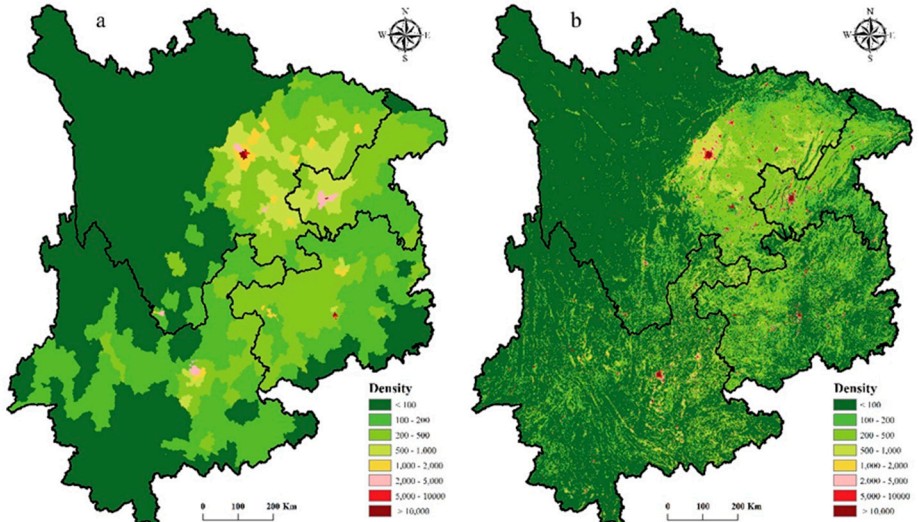

**Figure 4.** Comparison of the population distribution before and after spatializing in the study area: (**a**) county-level population density by census data; (**b**) the results of the spatial distribution of the population (SSDP).

### 4.2. Evaluation at both County and Town Levels

Accuracy assessment has always been difficult in population spatialization research. In general, the following methods are commonly used for assessment analysis: (a) comparing the results with previous studies; (b) using the data with finer statistical units for contrast verification; and (c) sampling verification in a field study. Considering the availability and operability of the data, we adopted a combination of those methods to assess the accuracy of SSDP; therefore, we chose to use the GPWv4, LandScan, and CGPD in 2010 for validation contrast at the county and township scales individually. The census data were treated as the true values. A previous study presented some conclusions about the comparison between population grid products which were used for contrast [5], and the assessment of SSDP was as follows.

At the county scale, owing to the adjustment of the population in SSDP, the percentage error was close to zero. In addition, to show the broad-scale patterns of error, a scatter plot with a box was applied (Figure 5). The relative errors (REs) with signs of towns in each dataset were expressed by different dots. Short horizontal lines on both ends indicate the minimum and maximum values, and

intersecting lines mark the 1% and 99% percentiles. Figure 5 shows that the range of MEs was −0.2 to 1.38 for GPWv4, −0.6 to 0.9 for LandScan, and −0.8 to 0.7 for CGPD, and outliers were uniformly distributed in the low and high-value areas. Although an adjustment method was applied to the three datasets, the fluctuation of error indicated that the adjustment by county census data in SSDP was effective and necessary.

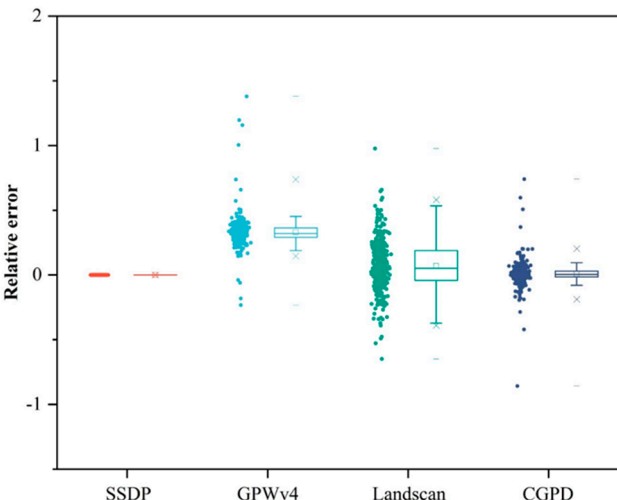

**Figure 5.** The scatter plot of relative error in 435 counties of four population grids.

Towns are the fourth level of administrative division in China and have a mean spatial resolution of 13.4 km [5]. Based on the generation of random numbers, we selected 500 towns. At the town scale, the 500 randomly chosen towns were used for the evaluation based on the accessibility and integrality of the township boundary and census data; then, the REs were calculated between the estimated population and census data in each town. Table 5 shows the accuracy justification indices based on the ME, MRE, RMSE, and MPE in these datasets at the township scale. The ME was 38.42% for GPWv4, 22.68% for LandScan, and 31.66% for CGPD, with MRE values of 48.51%, 57.25%, and 55.52% attained, respectively. When using the s-GWR model to estimate the population size, remarkable decreases in ME (7.59%) and MRE (33.63%) were attained. Meanwhile, this action was repeated by RMSE and MPE. The RMSE can reflect the deviation between the predicted and actual data, while the MPE commonly stands for the central tendency of data. The LandScan analysis showed a similar degree of dispersion (29,187) to CGPD, when compared to 25,161 of GPWv4, and both were significantly higher than the value of 17,381 for SSDP. The MPEs in ascending order were 0.3% for SSDP, 7.27% for LandScan, 17.13% for CGPD, and 30.68% for GPWv4. The results show that the total population was overestimated using these four products to different degrees, and this may have been caused by the misallocation of the population to incorrect areas in underdeveloped ones with sparse and scattered human settlements. The SSDP was expected to provide better precision and fewer errors. Figure 6 shows the correlations between the predicted population and census data in the 500 towns selected. Each point stands for the predicted population value and the corresponding statistical population value at the township level. We can see that the results of SSDP are more linear and concentrated than the other three in the relationship between the estimated population and census data. Moreover, when compared with the correlation coefficients of GPWv4 (0.739), LandScan (0.717), and CGPD (0.56), the SSDP had the highest value (0.763) between the estimated and statistical population.

**Table 5.** Comparison of accuracy in four population grids.

| Index | SSDP | GPWv4 | LandScan | CGPD |
|---|---|---|---|---|
| ME | 0.075 | 0.384 | 0.226 | 0.316 |
| MRE | 0.336 | 0.485 | 0.572 | 0.555 |
| RMSE | 17,381.578 | 25,161.567 | 29,187.063 | 29,187.083 |
| MPE | 0.239 | 0.35 | 0.419 | 0.356 |

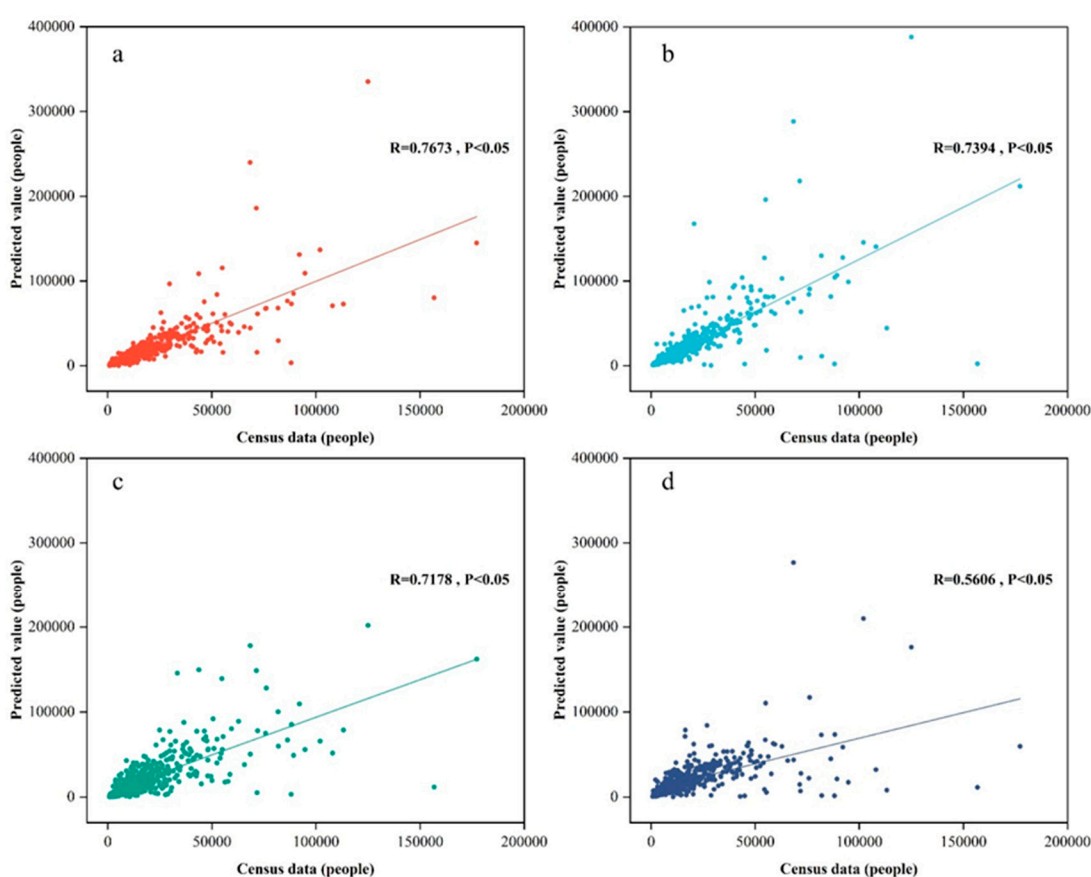

**Figure 6.** Comparison of the correlations between predicted populations with census data in 500 selected towns: (**a**) spatial distribution of the population (SSDP), (**b**) Gridded Population of the World (GPW)v4, (**c**) LandScan, and (**d**) China Gridded Population Datasets (CGPD).

To straightforwardly expose the error structure of these datasets in townships, we graded the relative error into five ranges and calculated the number of towns in different error ranges; this is illustrated as a bar graph (Figure 7). The towns whose ME values ranged from −20% to 20% were classified to be in an "accurately estimated" (AE) category due to having a relatively good estimation performance. Based on the ME values, the populations of the others were ordered as "extremely underestimated" (EUE; ≤−50%), "underestimated" (UE; −50% to −20%), "overestimated" (OE; 20% to 50%), or "extremely overestimated" (EOE; ≥50%). From EUE to EOE, the percentages of total samples that fit into those five ranges were 7.01%, 18.63%, 44.08%, 16.63%, and 13.62% for SSPD; 2.4%, 7.21%, 29.05%, 32.06%, and 29.25% for GPWv4; 14.82%, 18.43%, 24.24%, 14.62%, and 27.85% for LandScan; and 8.41%, 14.02%, 29.45%, 20.04%, and 28.05% for CGPD, respectively. As mentioned above, the population sizes were overestimated in the datasets, as it can be seen that the percentages of towns with OE populations were 61.32% for GPWv4, 42.47% for LandScan, and 48.09% for CGPD, which are much higher than their respective rates of UE (9.61%, 33.25%, and 22.44%); however, the SSPD had better performance in OE (30.26%) townships. The results show that overestimated populations of the GPWv4, LandScan, and CGPD were clustered mainly on the eastern Qinghai–Tibet Plateau,

northwest Yunnan Province, and Yunnan–Guizhou Plateau, which are all sparsely populated across wide areas. As for the SSPD, there were less overestimated towns and there were more dispersed. The percentage of BE townships in the SSDP increased to 44.08%, which was the highest value among these four datasets. Therefore, the s-GWR model was effective for decreasing errors and increasing precision, including reducing the rate of overestimated populations and improving the accuracy of estimated populations.

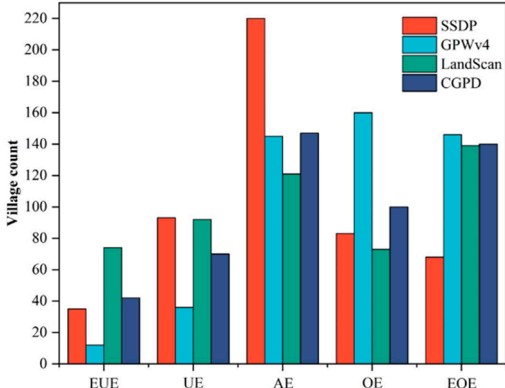

**Figure 7.** Village count with different error ranges in samples.

Meanwhile, we also used a scatter plot with boxes to show the details of errors in townships (Figure 8). Figure 8 shows that the REs of SSPD were more concentrated around zero, as the standard deviation of SSPD was 0.47 when compared with the values of GPWv4 (0.71), LandScan (0.79), and CGPD (0.84). Whether using the mean or median error, the SSDP performed better than the others. Additionally, the box plots showed that the range of MEs was −0.9 to 2.5 for SSDP, −0.9 to 8.4 for GPWv4, −0.9 to 4.3 for LandScan, and −0.9 to 8.3 for CGPD; this indicated that these datasets were skewed. No obvious differences were observed in the low-value area. When combined with the scatter plot analysis, we can see that the outliers were mainly distributed in the high-value area and were triggered by overestimation; SSDP attained a good result. Simultaneously, the variation in the population distribution between the county and township scales helped to clarify that the global model that did not consider the non-stationarity of variables is probably not suitable for the local area. In summary, not only was there minimal integrated error attained from the SSDP data derived by s-GWR, but there was also a minimum deviation error for predictions; this indicates that the SSDP conforms better to the actual distribution of the population than the other three products.

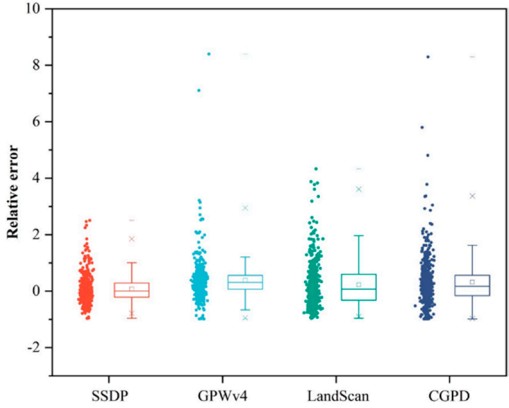

**Figure 8.** The scatter plot of the relative error in 500 towns from four population grids.

## 5. Conclusions

A population grid can directly reflect the range and intensity of human activities and is an effective indicator that provides necessary data for the study and characterization of the relationship between humans and the environment (climate, land cover, air pollution, etc.). In recent decades, downscaling has been widely used with the census population data to create grid level data using fusions with auxiliary data.

With the gradual deepening of the study of population spatialization, a wealth of research achievements have been attained based on night-time light data, but some problems still remain. Existing methods often ignore details that show the spatial variation of a population, perhaps weakening the accuracy of those methods. This paper discusses a population spatialization method based on s-GWR that can take into account the incorporation of the spatial stationarity of coefficients. The local and global patterns of variables were mixed when comparing this method to traditional global and local models to improve the accuracy of population mapping by representing the density variation of a population that varies with the geospatial position. Then, we estimated the population size by using the performance advantages of night-time light and land use data. Based on the correlation coefficients, the relationships between population and land use were described. Five land use classes (paddy fields, non-irrigated fields, urban areas, rural residential areas, and other built-up areas), which had significant positive correlations with the population, were selected. Then, we developed three indices (LE, NL, and NU) for each type of built-up area by combining these with DMSP/OLS data. In addition, two indices were developed based on the area of farmland considering the land use map error and the scattered population distribution in non-urbanized areas. Local and global factors were distinguished by a geographic variability test. Then, both local and global factors were fused by s-GWR in a mixed population model. Based on the eleven variables obtained, the adjusted $R^2$ of s-GWR reached 0.9, and the AICc value was 11,315, which is an increase by as much as 14% and a decrease by as much as 254, respectively, when compared with the pure global and local models. The results indicate that the s-GWR model can attain higher accuracy and stronger explanatory power than traditional models by synthetically considering the incorporation of the spatial stationarity of variables.

Additionally, by adopting an s-GWR model, we developed a map showing the spatial distribution of the population in the study area within four provinces in the southwest area of China in 2010 with a 1 km resolution. To show the accuracy assessment intuitively, three commonly used population datasets were used for comparison at the county and township levels along with census data, which mainly included four precision indices (ME, MRE, RMSE, and MPE). The error analysis indicated that the SSPD had improved significantly in terms of estimating accuracy controls when compared with the other three datasets and when the spatial variability of population density was considered. Compared with other models, the ME of SSPD decreased by up to 31%, MRE decreased by 24%, RMSE by 67%, and MPE by 12% at the township scale. Then, the grading statistics of the relative error at the township level showed that the accurately estimated towns of SSDP accounted for 44.08% from a total of 500 towns, and 30.26% and 25.62% of the towns were over and under-estimated, respectively. Compared to GPWv4, LandScan, and CGPD, SSPD performed better for accurate and overestimated towns, especially in terms of correcting the skewness characteristic of overestimated populations that existed in all four datasets and verified the results at the county scale. Furthermore, the overall details of the error distribution indicated that the relative error of SSPD had a smaller range and was concentrated around zero at the county level. The standard deviation of SSPD was 0.47, which was obviously lower than that of GPWv4 (0.71), LandScan (0.79), and CGPD (0.84). By combining these results with the results of the correlation test between estimated population and census data, it was also proven that SSPD derived by the s-GWR model has an advantage over other datasets in redistributing the population accurately.

The results of this paper confirm that when compared with traditional global or local models, the s-GWR method that considers the incorporation of the spatial stationarity of variables performs better, and SSDP has a higher accuracy than other datasets in the study area. This method can provide

valuable information for correlative research studies, such as studies of disaster evaluation, urban planning, and ecological assessment. Moreover, night-time light and land use data, which are all freely available on a global scale, are, therefore, much more suitable for population spatialization on a large scale where detailed data are lacking. When applied in SSDP, the low-cost process can be replicated easily. Nevertheless, there were several problems in this long-term study: (1) The limited 1 km resolution of DMSP/OLS data and land use data may have restricted their application on a small scale. Therefore, future research should consider utilizing finer resolution data, such as information related to buildings and new night time light, like the Luojia 1-01 satellite, to conform to the tendency of a higher pixel resolution level. (2) For sparsely populated areas, the population distribution datasets were also shown to overestimate the population, because the population distribution was affected by the collective influence of various factors, although the SSPD provided improved results when compared with other datasets. (3) The resampling also affected the data due to smoothing, especially for the areas with a sharp quantity change in population. (4) Before establishing an effective s-GWR model, a key step is to choose an effective criterion of divisional stationarity and non-stationarity. Therefore, before a perfect model that is widely used can be realized, population spatialization still needs further study.

**Author Contributions:** Conceptualization, Junnan Xiong and Kun Li; funding acquisition, Junnan Xiong; methodology, Junnan Xiong; project administration, Weiming Cheng; supervision, Junnan Xiong; validation, Junnan Xiong and Kun Li; writing—original draft, Kun Li; writing—reviewing and editing, Chongchong Ye and Hao Zhang.

**Funding:** This research was funded by the Strategic Priority Research Program of Chinese Academy of Sciences (XDA20030302), The Science and Technology Project of Xizang Autonomous Region (grant number XZ201901-GA-07), the Open Subject of Big Data Institute of Digital Natural Disaster Monitoring in Fujian (NDMBD2018003), the Southwest Petroleum University of Science and Technology Innovation Team Projects (2017CXTD09), and the National Flash Flood Investigation and Evaluation Project (SHZH-IWHR-57).

**Acknowledgments:** The authors also would like to appreciate the valuable comments from anonymous reviewers.

**Conflicts of Interest:** The authors declare no conflict of interest.

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
