# Peer review of "A Method of Population Spatialization Considering Parametric Spatial Stationarity: Case Study of the Southwestern Area of China"

_ijgi, doi:10.3390/ijgi8110495_

Round 1

Reviewer 1 Report

In my opinion, this is an interesting work where the authors apply spatial regression techniques to downscale and estimate population density. However, I must share that the final cartographic product is too coarse and that it would be interesting to replicate the same analysis to a finer spatial resolution to access if this methodology is feasible and if the results and validation would be accordingly. The tendency to map population density even in such large areas is to downscale to a higher resolution pixel level.

You may find some comments bellow:

Title: At a first insight, the title is not appealing for the reader; at the same extent is not very clear; authors may want to review the title of the manuscript.

References: check ref 57

Text:

L 80- What the authors mean by “very results”? What the authors state is true, but the quality of the data itself is more important than the choice of the algorithm.

L81- I believe it might be a contradiction when authors say that “Commonly used methods mainly include dasymetric mapping methods [28,29], spatial regression…” and in the abstract we can found that “However, little attention has been given to the spatial stationarity of variables models”. Additionally, in L94 ” Some studies have employed geographically weighted regression”, which in this case has less contradiction as GWR is only one of the spatial regression models. Nevertheless, It seems three different positions concerning the same idea. Spatial regression deals with the spatial stationarity.

L 242 – This section has too much information in my opinion. If summarized, it would be clear. Many studies already describe those theories and equations. Therefore, their explanation can be shorted and direct.

L 301- It must be shared why the authors choose those accuracy tests. Please, cite other works where those validation tests were applied, if possible, within the same study field.

Others:

Authors need to reflect in which extend is dasymetric mapping a traditional method.

Why the authors did not want to produce coefficients maps to spatially visualize the regional variation? The areas with a strong relationship between the independent variable and dependent could be spotted.

Same question regarding the R2.

Did the authors consider other essential validation tests such as the Koenker or Jacque-Bera in their analysis?

GWR should be considered part of an ESDA because it can create and assume false spatial patterns. Therefore, authors need to intake special attention to that.

Even if many studies addressed GWR limitations, authors need to identify s-GWR limitations and weaknesses.

Reviewer 2 Report

I have attached a document with my review.

Reviewer 3 Report

The current manuscript is good as an idea and is very well presented. 

However, the paper needs to be proofread for the English Language. Moreover, tables should be aligned properly according to the paper size.
